# Hybrid UAV Hazard Detection Approach based on Open-Vocabulary Detection and Mivar Expert System

**Oleg Lamcev, Ivan Chernyadiev, Aleksandra Maksimova**
Laboratory of Intellegent Systems
Institute of Applied Mathematics and Mechanics
Donetsk, Russian Federation
`gelo2003@yandex.ru`, {`chernyadev-i,maximova.alexandra`}`@mail.ru`

## Abstract

The integration of neural network methods in computer vision with logical inference based on a Mivar expert system allows leveraging the advantages of both paradigms: high efficiency in processing unstructured visual data and the interpretability of decisions made based on formalized rules. An analysis of various computer vision tasks was conducted, demonstrating that OVD (Open-Vocabulary Detection) is the preferred tool for dynamic rescue operation scenarios. OVD provides the best balance between the flexibility of detecting arbitrary categories, reliability when working with multiple objects, and the availability of data for training. Modern vision-language architectures, their features, and advantages were investigated. YOLO-World was selected as the base model, as it best meets the stringent requirements of real-time operation, achieving high processing speed while maintaining the flexibility of an open vocabulary. A fine-tuning procedure for the model was carried out, which included freezing the text encoder and the early layers of the convolutional backbone, as well as combining the Flickr30k, VisDrone, and SARD2 datasets. Using Flickr30k helped preserve the quality of the vision-language space, while the specialized datasets adapted the model to real-world application conditions. The fine-tuned model showed a significant increase in accuracy (mAP@50 rose from 0.0974 to 0.342, and mAP@50:95 from 0.0673 to 0.202), and also gained the ability to correctly recognize human poses specific to rescue operations and types of vehicles in drone imagery. A system of parameters and rules for the Mivar Expert System (MES) is proposed, which allows transitioning from simply listing objects in an image to a comprehensive situation assessment. This transforms the system into an active operator assistant, capable not only of detecting but also of interpreting threats. Thus, the developed hybrid intelligent system, combining the YOLO-World detector and the Mivar expert system, fully meets the stated goal and specified requirements:

- Real-time operation due to the optimized architecture;
- Flexibility of control through text prompts in natural language;
- Interpretability and logical validity of decisions thanks to Mivar logical inference.

## 1 Introduction

Today's most effective computer vision systems are based on the use of neural network models. The key advantage of such models is their efficient processing of unstructured data and automatic extraction of significant features without manual feature engineering. Traditional machine learning methods are heavily dependent on human data preprocessing. Computer vision algorithms based on neural networks deliver high performance; however, their results are poorly interpretable and prone to hallucinations.

Models built on Multidimensional Informational Variating Adaptive Reality (MIVAR) (Varlamov, 2011) are interpretable. They constitute an expert system that provides logically sound analysis and interpretable decisions. The drawback of the Mivar expert system (MES) is its strictly formalized structure and inability to work with raw data independently. In recent years, a hybrid approach has been gaining popularity. It combines the advantages of both neural network and Mivar methods: feature extraction by the neural network and logical inference using Mivar rules.

Typically, the neural network detector in hybrid systems has a fixed set of classes, making it insufficiently flexible. This poses a problem for dynamic scenarios, such as detecting dangerous situations using unmanned aerial vehicles (UAVs) in rescue operations. The emergence of new relevant objects requires labor-intensive retraining of the model.

This paper proposes a novel hybrid approach that combines a MES with an open-vocabulary object detector. Unlike closed-set models, such a system compares visual and textual features within a unified vector space, expanding capabilities by detecting previously unknown objects using text prompts. This allows the UAV operator to dynamically direct the model's attention to arbitrary objects not included in the training dataset, making this method promising for real-world applications in rapidly changing scenarios.

The aim of this work is to develop a hybrid intelligent system that combines a neural network-based computer vision module and a Mivar expert system. The system is designed for real-time detection of dangerous situations during rescue operations using data obtained from UAVs.

The following key requirements are considered during the system's development:

- Specifics of input data. Images and video streams come from a UAV camera, which determines a top-down or oblique shooting angle, potential dynamic distortions, the influence of lighting and weather conditions, and limited resolution. For fine-tuning the neural network and forming the training set, datasets containing aerial imagery or data captured from drones must be used. This is necessary to ensure adequate generalization capability of the model in real-world operating conditions;
- Real-time operation. The task is dynamic in nature, where the speed of decision-making is critical. The system must process the video stream or sequence of images at a frequency sufficient for prompt response (at least 10-30 frames per second, depending on available computing resources). To meet this requirement, optimized neural network architectures and other inference acceleration methods should be used;
- System control via text prompts in natural language. The UAV operator must be able to specify target object classes or situations to the system through textual descriptions. This functionality allows for the detection of objects and situations not present in the training dataset.

These requirements determine the architectural and algorithmic solutions of the hybrid system, aimed at achieving high efficiency, adaptability, and reliability in real-world rescue operation conditions.

## 2 BACKGROUND AND RELATED WORK

The problem is addressed using computer vision methods based on neural networks, combined with a logical approach. Let's consider existing neural network approaches.

Object Detection (OD) is one of the fundamental tasks in computer vision. It involves finding and localizing individual objects within an image or video stream, followed by assigning each detected object a class from a pre-defined, fixed set of categories. The advancement of multimodal approaches has led to the emergence of new object detection paradigms that extend classical detection methods by incorporating natural language descriptions. Let's look at different problem options.

Phrase Grounding (also known as Referring Expression Comprehension or Visual Grounding) is the task of localizing a single object in an image based on a detailed textual description. The model receives an image and a text query that describes one specific object in detail, and must return a bounding box for that object. It is assumed that the described object is definitely present in the image.

Open-Vocabulary Detection (OVD) is an extension of classical object detection where the model is capable of detecting objects from arbitrary categories, specified by short text descriptions in the form of a list of class names. Unlike traditional OD, OVD is not limited to a fixed set of categories defined during the training phase.

Described Object Detection (DOD) (Xie et al., 2023) is a more general and flexible generalization of the REC and OVD tasks. DOD allows the use of language expressions of arbitrary length and complexity: from short category names to detailed descriptions of an object's appearance, context, or attributes. The model must correctly handle cases where the described object is absent from the image, avoiding false positive detections.

## 2.1 The OVD approach selection

In the formulation of the OVD task, the list of detectable objects is expanded to a list with arbitrary short category names, but it does not account for more complex language descriptions. The REC task, on the other hand, focuses on the precise localization of a single target based on a detailed expression but assumes the target's presence in the image. In real-world scenarios, this assumption is often violated, leading to false positive predictions. The DOD task supports language queries of any complexity, from simple category names to long and detailed descriptions and correctly handles cases where described objects are absent from the image by suppressing false positives.

This work focuses on the OVD task. Compared to REC, which is aimed at localizing a single object based on a unique description, OVD is more effective in scenarios with multiple objects of the same category, supports the absence of objects, does not require unique and potentially ambiguous descriptions, and is closer to classical object detection tasks. Unlike DOD, which supports complex descriptions with attributes and relations, OVD is simpler and more accurate for standard categories, avoids accuracy drops due to long descriptions, and benefits from more mature methods, established benchmarks, and better performance on typical open-vocabulary tasks.

The key advantage of OVD remains the ease of data access: standard object detection datasets (COCO, LVIS, Objects365) are used directly, whereas DOD requires rare and expensive annotations with detailed descriptions, often necessitating complex adaptation or synthesis.

Thus, OVD provides an optimal balance between open-vocabulary flexibility and reliability in detecting multiple objects, requiring significantly lower data preparation costs. It is precisely these advantages that make OVD the preferred approach for dynamic hazard detection systems using UAVs, where it is necessary to promptly detect unauthorized drones, suspicious objects, or multiple threats simultaneously in protected areas, ensuring high response speed, resilience to new types of threats, and minimizing false positives in critical security and monitoring scenarios.

## 2.2 The role of Vision-Language Models (VLMs) in the OVD task

At the core of OVD lies the use of powerful multimodal models, such as CLIP (Radford et al., 2021) and its analogs. CLIP is trained on vast amounts of image-text pairs using contrastive learning: the model brings the vector embeddings of corresponding pairs closer together and pushes apart those of non-corresponding pairs. This results in the formation of a shared multimodal vector space where visual features are aligned with their textual descriptions for an enhanced visual-semantic representation. This enables the model to generalize to new categories not seen during training, as it can match arbitrary text prompts (class names, descriptive phrases) with visual features.

However, directly applying CLIP to the object detection task encounters a fundamental modality gap. Despite its central role, models like CLIP themselves are not capable of object localization-they only provide a global similarity score. CLIP is trained at the whole-image level, matching it with a global description, whereas detection requires localized understanding. This discrepancy between image-level pre-training and the need for region-level analysis has led to the development of specialized OVD architectures that bridge this gap.

Modern OVD models typically consist of three key components. First, a visual encoder extracts visual features from the input image. While early work relied on convolutional networks (e.g., ResNet), architectures based on transformers, such as the Vision Transformer (ViT) or Swin Trans-

former, now dominate. These divide the image into patches, embed them, and process them sequentially, obtaining a dense feature grid where each vector represents a local area.

Second, a text encoder (usually transformer-based, as in CLIP or BERT (Devlin et al., 2019)) encodes the user's text prompt-from simple class names to complex descriptions-into a high-dimensional semantic representation.

Third, a modality fusion mechanism integrates the visual and textual features. Early approaches simply concatenated vectors from different modalities, but modern architectures use more sophisticated methods, particularly cross-attention. In this mechanism, visual features serve as queries for the textual features, allowing the model to dynamically weight and combine information.

The diversity of OVD architectures reflects different strategies for overcoming the modality gap. The choice of a specific implementation determines the balance between accuracy, speed, and generalization capability. Therefore, an analysis of various OVD models will be conducted in the next section.

## 3 ANALYSIS OF EXISTING VLMS FOR SOLVING OVD

### 3.1 FLORENCE-2

The Florence-2 model (Xiao et al., 2023), developed by Microsoft researchers, offers a unified approach to solving a wide range of image processing and vision-language interaction tasks based on a single architecture.

The model takes an image and a task-specific text prompt as input, and generates a text response as output, which contains the boundaries of the detected object when necessary. This allows it to solve a broad spectrum of tasks (caption generation, object detection, referring expression comprehension, segmentation, OCR, etc.) using a single architecture and a single set of weights.

The model's architecture includes several key components. The DaViT (Data-efficient Vision Transformer) visual encoder transforms the input image into a sequence of visual representations, effectively capturing spatial and semantic information at different hierarchical levels. The resulting visual representations are projected into a shared representation space and combined with textual representations, which are created from the prompt using an extended language tokenizer and an embedding layer (based on BART). The final multimodal sequence is processed by a transformer encoder-decoder, which generates the output sequence according to the task.

For tasks involving spatial localization, special location tokens representing quantized coordinates have been added to the vocabulary. Supported formats include: rectangle (x0, y0, x1, y1), quadrilateral, and polygon.

Florence-2 was trained in a multi-task learning regime on the FLD-5B dataset (126 million images, 5.4 billion annotations). It embodies the idea of a foundation model for computer vision, similar to trends in natural language processing, where a single architecture and a unified way of representing tasks ensure high knowledge transferability and efficiency across a wide range of applications.

### 3.2 KIMI-VL

Kimi-VL (Team et al., 2025) is an open vision-language model built on a Mixture-of-Experts (MoE) architecture. It combines high performance in multimodal reasoning tasks with significantly reduced computational costs: with a total of around 16 billion parameters, only about 2.8 billion parameters of the language decoder are activated during inference (in the base Kimi-VL-A3B version).

The model's architecture includes three key components: the MoonViT visual encoder supporting native resolution; an MLP-based projector ensuring efficient alignment of visual and language representations; and an MoE language model that dynamically activates only a small fraction of its parameters. MoonViT, initialized from SigLIP-SO-400M and fine-tuned for high-resolution work, uses an image packing technique and 2D rotary position embeddings, allowing it to process images with resolutions up to several megapixels without the need for fixed-size reshaping.

Kimi-VL demonstrates strong results in complex multimodal tasks: multi-step agent interactions, OCR, image and video understanding, visual mathematical reasoning, and working with multiple images and long videos.

Despite its impressive characteristics, the model has natural limitations due to its scale. In tasks requiring very large volumes of pure linguistic knowledge or solving highly specialized niche problems, its performance lags behind larger models. Furthermore, even with an extended context window (up to 128K tokens), technical challenges related to processing extremely long sequences remain.

### 3.3 YOLO-WORLD

YOLO-World (Cheng et al., 2024) implements the "prompt-then-detect" paradigm, which completely eliminates the need for real-time text encoding. User text queries are pre-processed into an offline vocabulary of embeddings, significantly speeding up the inference process and greatly reducing the computational load.

Unlike traditional detectors limited to a fixed set of classes, as well as earlier open-vocabulary methods that use prompt encoding during runtime, YOLO-World combines high speed with flexibility. The detection vocabulary can be dynamically adapted to a specific task without compromising performance, making the model convenient for practical application in real-world scenarios.

The model's architecture includes a detector based on Ultralytics YOLOv8, which extracts multi-scale visual features from the image; a transformer-based text encoder pre-trained within the CLIP framework; and the RepVL-PAN network. The latter is responsible for multi-level cross-modal fusion of visual features and text embeddings. During inference, embeddings from the offline vocabulary are transformed into static weights for the RepVL-PAN network, completely eliminating the need for dynamic text-based computations during runtime.

Three model size variants exist. The small version (YOLO-World-S) has 13 million parameters during inference (77 million in training mode), the medium version (YOLO-World-M) has 29 million parameters (92 million in training), and the large version (YOLO-World-L) has 48 million parameters (110 million in training).

Compared to previous OVD methods, YOLO-World demonstrates a significantly better combination of speed and accuracy. The proposed paradigm and architectural solutions make the model a powerful and practically applicable tool for tasks requiring flexible, fast, and resource-efficient object detection in open-vocabulary scenarios.

## 4 MIVAR SYSTEM AS LOGICAL PART FOR DECISION MAKING

The Mivar approach represents a modern methodology for developing intelligent systems, based on the use of Mivar knowledge bases and corresponding bipartite network structures. Foundational to this is a unique form of knowledge representation that combines elements of production systems, Petri nets, and entity-relationship models, creating a unified mathematical foundation for logical inference and information processing (Maksimova & Varlamov, 2011).

A key feature of the Mivar approach is its implementation of logical inference with linear computational complexity, even when working with knowledge bases containing a large number of rules and objects. This is achieved through the specialized organization of Mivar networks, in which parameters, rules, and connections are structured as a multidimensional directed graph with clearly separated layers of "entities" and "relations." This approach ensures high scalability, the possibility of efficient parallel processing, and resilience to the increasing complexity of the subject domain.

Mivar systems demonstrate several important advantages compared to traditional expert systems and some modern approaches based on neural networks:

- High adaptability and evolutionary expandability of knowledge;
- Ability to operate under conditions of incompleteness, uncertainty, and inconsistency in initial data;

- Support for rapid modification of system behavior without changing the base architecture or retraining models;

- The possibility of explainability of decisions, since inference is built on explicit rules and cause-and-effect relationships.

Mivar technologies have found application in a wide range of subject areas.

In robotics, they are used to build systems for intelligent action planning and autonomous control of complex manipulators and mobile robots. In medicine, Mivar expert systems are applied for differential diagnosis of diseases, forming personalized treatment protocols, and supporting clinical decision-making. In the transportation sector, the Mivar approach enables intelligent control of unmanned vehicles, route optimization, and dynamic planning in changing traffic conditions. Mivar solutions demonstrate significant potential in information security tasks (threat and anomaly analysis), industrial automation (production process management and predictive equipment maintenance), as well as in education-for creating adaptive learning systems, intelligent simulators, and electronic learning environments.

One of the most significant advantages of Mivar expert systems (MES) is their high flexibility when modifying operational logic without the need to redesign the architecture or retrain neural network components. This property is particularly important in dynamically changing subject areas, where requirements for interpreting events or actions may be adjusted during system operation.

Thus, the Mivar approach opens a promising direction for the development of logical artificial intelligence, combining high computational efficiency, explainability of decisions, and unprecedented flexibility in knowledge modification. These characteristics make Mivar technologies especially attractive for creating reliable, scalable, and adaptive intelligent systems in real-world conditions. Further research and practical implementations will allow for a fuller realization of the potential of this approach in solving complex interdisciplinary problems.

## 5   ADAPTATION OF THE NEURAL NETWORK TO THE SUBJECT DOMAIN

Adaptation of the Neural Network to the Subject Domain. The developed neural network module is based on a pre-trained open-vocabulary model, YOLO-World, in its minimal configuration. This model initially contains 77 million trainable parameters. After removing the text encoder, the text embeddings are transferred to fixed parameters of the classification head, and the number of active parameters during inference is reduced to 13 million. This allows achieving an operating speed of 75 frames per second.

The model accepts input data of two modalities:

- An RGB image obtained from the UAV camera;

- A text query entered by the UAV operator in natural language. These can describe both individual object classes ("person," "dog," "car") and their specific attributes and states ("person in a green T-shirt," "sitting dog," "black SUV"). This formulation enables flexible object detection without needing to retrain the model for each new class or attribute.

To adapt the pre-trained model to the target subject domain (aerial surveying from UAVs, primarily search and rescue tasks, and monitoring of urban/natural environments), a fine-tuning procedure is applied. During this process, the text encoder (based on CLIP) is frozen by default, and additionally, we freeze the first 10 layers of the YOLO model (the backbone network). This reduces the risk of catastrophic forgetting of general vision-language representations and decreases the required computational resources.

A combination of three datasets was used for fine-tuning. Flickr30k is a large dataset of images with detailed text descriptions and bounding boxes. This dataset is used to preserve and maintain the quality of the vision-language space achieved during the pre-training stage. Our own combined dataset, compiled from two open datasets focused on aerial surveying from UAVs: VisDrone and SARD2. This combination of datasets allows training the model to identify various types of vehicles and people in different poses, which is necessary for our subject domain.

The following approach was used for fine-tuning the model: fine-tuning was performed on our own combined dataset (10,609 images) for the selected subject domain. From these, 7,857 were allocated for training, 944 for validation, and 1,808 for testing. In these datasets, integer class labels are replaced with string class names to bring the data into a format compatible with the open-vocabulary paradigm. The Flickr30k dataset was used to connect textual and visual embeddings in a generalized vector space.

Thus, the combination of Flickr30k and the target datasets ensures a balance between preserving the model's generalization capability and its specialization for objects and scenes typical of unmanned reconnaissance and aerial monitoring tasks.

## 6 DESCRIPTION OF RESULTS AND METRICS

To evaluate the effectiveness of the neural network model, test images from the VisDrone and SARD2 datasets were used. Figure 1 presents two normalized confusion matrices: the matrix on the left corresponds to the pre-trained model, and the one on the right corresponds to the model fine-tuned on our custom dataset. The pre-trained model demonstrated metrics of 0.0974 (mAP@50) and 0.0673 (mAP@50:95). After fine-tuning, the model's performance significantly improved, reaching 0.342 and 0.202 on the respective metrics.

As a result of fine-tuning, the model not only increased detection accuracy but also became capable of recognizing people in various poses (walking, lying, sitting, standing) and types of vehicles in images obtained from UAVs. The output of the pre-trained and fine-tuned models is shown in Figure 2 and Figure 3.

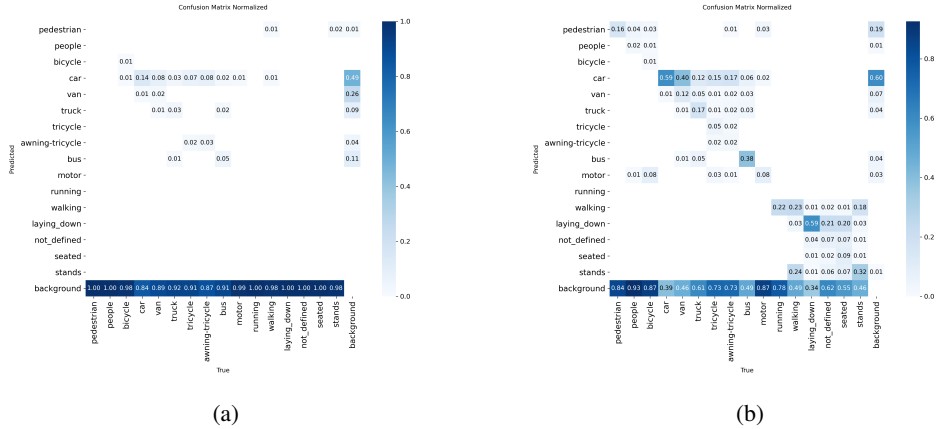

Figure 1: Confusion matrices of the pre-trained model (a) and the fine-tuned model (b).

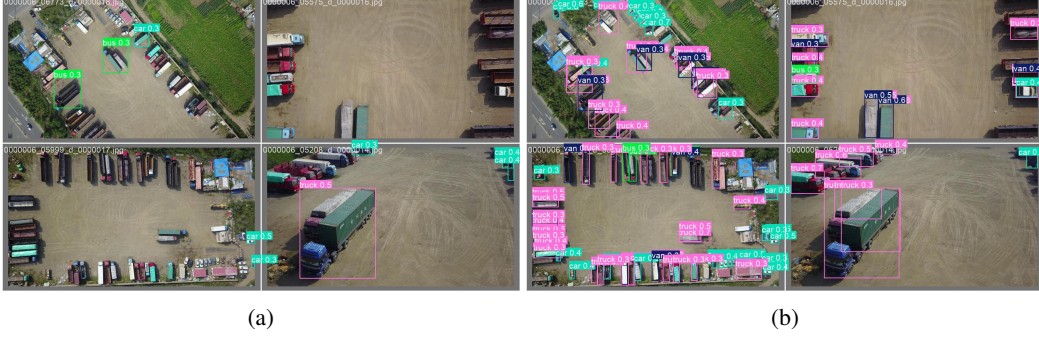

Figure 2: Output of the pre-trained model (a) and the fine-tuned model (b) for identifying types of vehicles.

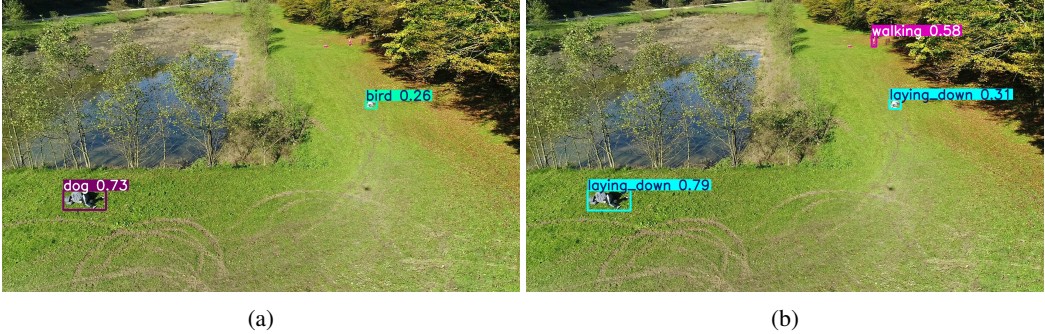

|             (a)             |             (b)             |

Figure 3: Output of the pre-trained model (a) and the fine-tuned model (b) for determining human poses.

## 7 CONCLUSIONS

The hybrid system, combining the MES with an open-vocabulary object detector, opens a new path for intelligent support of rescue operations using UAVs. In the future, such a paradigm could become the foundation for universal intelligent platforms that automatically adapt to new types of threats and scenarios through dynamic expansion of the vocabulary and rules. This is particularly valuable in conditions of uncertainty, contradictory data, and rapidly changing environments.

Further research may be directed towards significantly expanding the base of Mivar rules to cover natural disasters, technogenic accidents, and complex multifactor emergency situations. Integrating the system with UAV telemetry (flight altitude, GPS coordinates, heading) and camera orientation appears to be a promising direction, as it would enable a comprehensive spatiotemporal analysis of the observed scene. Furthermore, the developed architecture has high potential for adaptation to tasks in information security, industrial monitoring of critical infrastructure, and autonomous robotic systems, where high response speed, flexibility, and complete transparency of decisions made are simultaneously required.

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
