# OpenReview forum: "Hybrid UAV hazard detection approach based on Open-Vocabulary Detection and Mivar Expert System"
_mathai.club/MathAI/2026/Conference — 2026 Oral_

### Official Review · Reviewer_u17m · 2026-03-11
**The Abstract “Hybrid UAV hazard detection approach based on Open-Vocabulary Detection and Mivar Expert System” submitted to MathAI 2026 Conference  describes the hybrid approach for localization of new objects using an open-vocabulary detector and the interpretability of mivar expert systems, solving the problem of their limited flexibility in dynamic conditions.**

**Rating:** 7
**Confidence:** 3

**Review:**

The Abstract “Hybrid UAV hazard detection approach based on Open-Vocabulary Detection and Mivar Expert System” submitted to MathAI 2026 Conference.  It describes the hybrid approach for localization of new objects using an open-vocabulary detector and the interpretability of mivar expert systems, solving the problem of their limited flexibility in dynamic conditions. The Abstract is well -written and present valuable results for readers. Classical machine learning methods  depend on extensive manual engineering and preprocessing. While neural network-based computer vision algorithms deliver high performance, their results are difficult to interpret. Interpretable models are those built on the Multidimensional Information Variable Adaptive Reality (MIVAR). A hybrid approach combines the advantages of neural network and MIVAR method. The neural network detector in hybrid systems has a fixed set of classes, making it inflexible. A MIVAR expert system with an open-vocabulary object detector helps to operator to dynamically direct the model's attention to arbitrary objects not included in the training set. Open-Vocabulary detector based on a combined VisDrone and SARD2 dataset (images of people and vehicles captured from a drone's flight altitude) and a MIVAR module with a set of logical rules MIVAR module analyzes the detector's output and makes decisions about the presence of dangerous situations. The proposed system retains the advantages of the hybrid approach and expands them by detecting previously unknown objects. This makes the approach promising for real-world applications in rapidly changing scenarios.

---

### Official Review · Reviewer_nBaQ · 2026-03-12
**Brief Summary of Review for "Hybrid UAV hazard detection based on Mivar expert system and open-vocabulary object detector"  The paper proposes a hybrid system combining YOLO-World with a Mivar expert system for UAV-based rescue operations. While the topic is relevant, the work has significant flaws: mAP metrics remain too low for practical use (0.342/0.202), the Mivar integration is superficially described without technical details, and novelty is limited—fine-tuning existing models is engineering, not science. No comparison with baselines or justification for Mivar over simpler rule-based approaches is provided. Major revisions required.**

**Rating:** 4
**Confidence:** 4

**Review:**

Review
1. Assessment of the quality, clarity, originality, and scientific significance of the work
Quality. The analysis has limited depth: the metrics mAP@50 = 0.342 and mAP@50:95 = 0.202 after fine-tuning remain at a level insufficient for practical application in critical scenarios (rescue operations, emergencies). For comparison, modern detectors in the aerial survey domain achieve mAP@50 values > 0.5. If the detectors discussed in the article allow solving previously inaccessible tasks (which might justify the modest mAP@50 and mAP@50:95 scores), this should be explicitly stated in the text, which was not done.
Clarity. The text of the article is well-structured into 7 sections, two of which have subsections, but it contains many well-known (correct) statements. However, the question of why the Mivar approach is preferred over other methods of creating expert systems is left unaddressed. The section dedicated to the integration of the neural network module with the Mivar system is written superficially, making it difficult for the reader to understand how these two parts technically interact.
Originality. The originality of the "YOLO-World + Mivar" hybrid approach is relative. The very idea of combining a neural network detector with an expert system is not new and dates back decades (work on hybrid intelligence from the 80s and 90s). The newest elements are the use of a Mivar system in combination with an open-vocabulary detector, but these components are already well-known individually. Other authors simply have not considered the issue of combining an open-vocabulary detector, as implemented in YOLO-World. The authors' contribution boils down to fine-tuning on standard datasets, which, by modern standards, is more of an engineering task than a scientific achievement.
Scientific significance. Essentially, the article describes the trivial integration of existing open components (YOLO-World, Mivar) with minimal adaptation for a specific task. This resembles a report on applied development more than a description of the results of scientific research.
•	The work is applied and demonstrative in nature. There is no theoretical justification for integrating VLM vector representations with Mivar logical rules.
•	It is not shown that the proposed system solves problems of interpretability: Mivar rules may be transparent, but the input data for them (the detector's output) remains a "black box."
2. Description of main ideas contained in the article.
The main ideas contained in the article can be summarized as follows:
Integration of an Open-Vocabulary detector (YOLO-World) with the Mivar expert system (MES). The authors propose using text prompts to dynamically expand the classes of detectable objects without retraining the model. The idea is sound, but it reproduces the content of the original YOLO-World work, and the authors do not describe what constitutes novelty concerning that cited article.
A system of Mivar rules for situational assessment. It is assumed that the Mivar system will analyze the detection results and make decisions about the presence of dangerous situations. Instead of simple object detection, the system should interpret their combinations and context (e.g., if "a person is lying next to a damaged car" → "medical assistance required"), forming an explainable decision for the UAV operator.
Fine-tuning YOLO-World on a combined dataset (VisDrone + SARD2 + Flickr30k). A procedure is proposed for adapting the pre-trained model to the aerial survey domain by freezing the text encoder and the early layers of the backbone network, which should preserve general visual-linguistic representations while specializing in target objects.
All three ideas are correctly formulated, but their implementation in the article is not supported by descriptions or evidence of the effectiveness of the authors' chosen implementation.
3. Degree of novelty, relevance, and significance of the claims
The novelty of the claims is questionable. Points 1 and 3 are a direct reproduction of ideas from the works (Cheng et al., 2024, referenced in the bibliography) and standard practices of fine-tuning neural networks. The concept of open-vocabulary detection and "neural network + expert system" hybrid architectures is widely known. The main claim to novelty—hybridization with Mivar—is not supported by comparisons with other hybrid systems (which are very popular now), making it impossible to assess whether this is truly a new contribution.
The relevance of the topic (rescue operations with UAVs) is undeniable; however, the usefulness of the proposed solution is not justified by comparison with similar systems. There are simpler and more effective ways to solve the problem that do not require a cumbersome two-component architecture. For example, one could fine-tune YOLO-World not only for object detection but also for situation classification, or use a set of prompts like "person next to fire" directly in the open-vocabulary detector, eliminating the MES.
The significance of the claims is not supported by convincing evidence. The authors state that the MES provides "explainability of decisions" and "flexibility in modifying logic," but do not provide a single example of how this is implemented or why it is better than simply programming rules in a common language (e.g., Python). Without comparison with similar systems, the use of Mivar appears as an artificial add-on. The claim of "real-time" performance (75 FPS) is not supported by tests on target UAV hardware (e.g., NVIDIA Jetson). The lack of analysis of robustness to noise, scale changes, and occlusions calls into question the applicability of the system in real-world conditions.
Conclusion: An urgent problem is being addressed with an insufficiently novel and poorly substantiated method.
4. Assessment of compliance with the requirements for logic of presentation, formatting, and terminology
Logic of presentation. At the macro level, the logic of presentation is followed (introduction-review-proposal-experiment). However, it is disrupted when describing the central issues. The most important question—how exactly the neural network transmits data to the MES and how the MES makes decisions—remains unanswered. Section 2 (Background) contains an excessive review of models (Florence-2, Kimi-VL) not used in the experiment, which dilutes the focus when perceiving the article's material. The transition from the analysis of VLMs to the description of Mivar (Section 4) is made without justifying why Mivar was chosen over other interpretable methods (e.g., neuro-symbolic approaches, decision trees). The logical connection between Section 5 (fine-tuning) and Section 4 (Mivar) is not obvious. It is unclear why a detailed description of YOLO-World fine-tuning was needed if it is not related to the operation of the MES and does not affect the logical inference.
Formatting. The text in the figures (Figs. 1–3) is very small—requiring significant magnification and even then, it is difficult to read. Formulas and metrics are presented without explaining the calculation procedure (e.g., how mAP was averaged across classes).
Terminology. Terms are used correctly, but the term "Mivar" is introduced without reference to a formal mathematical definition, making it difficult for readers unfamiliar with Varlamov's work to understand. In the sections on Mivar, the authors overuse specific terminology ("bipartite graphs," "multidimensional directed graph") without explaining it in the context of the task, which seems like an attempt to give weight to trivial things through pseudo-scientific language. The mixing of the concepts of "open-vocabulary detection" and "described object detection" without clear distinction introduces terminological confusion.
Formatting and terminology require refinement to meet academic standards.
5. Substantiated comments and recommendations for edits
Ensure interconnection between parts. Currently, the article falls into two weakly connected parts: "fine-tuning YOLO-World" and "description of the positive properties (without comparative analysis) of Mivar." A description of how these parts connect should be provided. Does the neural network output look like JSON with coordinates and classes, or something else? How does the MES receive this data? What does the MES knowledge base for rescue operations look like? It is desirable to answer the listed questions with examples: add an architecture diagram with data flow descriptions and a working example (e.g., "search for a victim in the forest"), show code or pseudocode for MES rules.
Insufficient elaboration of the Mivar module description: describe the rule verification procedure: who forms them, how consistency is checked; demonstrate how the system generates explanations for the operator; The authors have not shown that the MES is truly necessary. Any rules of the type "IF object = person AND object = fire, THEN danger" could be implemented with a simple Python script working with YOLO's output data. What advantage does complicating the system with a heavy expert shell provide?
Strengthen the description of experimental validation. Present an error analysis: confusion matrix by class, examples of false positives, PR curves. Provide an assessment of inference latency on target hardware (not just stating theoretical 75 FPS).
Lack of comparison. There is no comparison with baseline approaches. For example, how does the proposed system work compared to just YOLO-World, where prompts like "person, fire, smoke, dangerous situation" are passed? Perhaps the MES is not needed at all, and the problem is solved by direct prompting.
Formatting: Standardize terminology and add a glossary of abbreviations. Improve the readability of text in graphic figures.
Conclusion
Reject in its current form. The article addresses a relevant problem, but the presented solution is not supported by sufficient theoretical and experimental evidence, the novelty of the contribution is limited, and the formatting requires improvement. After substantial revision (adding comparative analysis, justification for choosing MES, refining the figures), the work may be reconsidered.

---

### Official Review · Reviewer_wLDQ · 2026-03-13

**Rating:** 5
**Confidence:** 3

**Review:**

This paper proposes a hybrid intelligent system for real-time hazard detection using UAVs in rescue operations. It combines a YOLO-World open-vocabulary detector, which allows operators to use natural language prompts to identify objects, with a Mivar Expert System (MES) for logical decision-making.

Strengths:

1. The combination of an open-vocabulary detector with a symbolic Mivar expert system is a timely and innovative solution to the real-world problem of dynamic hazard detection.
2. The authors provide a clear and logical rationale for their technical choices, particularly the selection of the OVD task and the YOLO-World model for its speed and efficiency.
3. The fine-tuning process for the YOLO-World model is well-executed and validated by a substantial improvement in quantitative results on the target aerial datasets.

Weaknesses:

1. The paper fails to demonstrate or evaluate the complete hybrid system. There are no examples or metrics showing how the Mivar system uses the detector's output to make decisions about hazardous situations.
2. The role of the MES is described only in general terms. The paper lacks crucial details, such as examples of the rule base, how neural network outputs are converted for logical processing, and the format of the final decision.
3. The evaluation is limited to object detection metrics (mAP). It does not assess the end-to-end performance of the system, such as hazard classification accuracy or full-system latency.

Conclusion:
This article presents a promising and well-motivated approach to a challenging problem. The work on adapting YOLO-World for aerial imagery is solid and valuable. However, the paper is incomplete, as it lacks any demonstration or evaluation of the core contribution—the integration of the detector with the Mivar expert system. To be a significant contribution, the paper must include a detailed case study and end-to-end evaluation of the complete hybrid system in action.

---

### Decision · Program_Chairs · 2026-03-14

**Decision:**

Accept (Oral)

**Comment:**

Dear Author(s),

On behalf of the Program Committee of the International Conference on Mathematics of Artificial Intelligence (MathAI 2026), we are pleased to inform you that your paper has been accepted for an oral presentation at MathAI 2026.

Your paper was evaluated through a rigorous two-stage review process involving both automated screening and expert review by members of the Program Committee. The reviewers recognized the quality and contribution of your work.

Presentation details:

- Format: Oral presentation (15–20 minutes + 5 minutes Q&A)
- Mode: You may present either in person (offline) at the conference venue in Sirius, Russia, or remotely via Zoom. Please indicate your preferred mode when confirming your participation.
- Conference dates: Marh 30 - April 3, 2026
- Website: https://mathai.club

Next steps:

1. Please confirm your participation and presentation mode by replying to this email mathai.club@yandex.ru no later than March 15, 2026 18:00 Moscow time.
2. If you plan to attend in person, the organizing committee will provide accommodation details separately.
3. Please prepare your final camera-ready manuscript according to the formatting guidelines available at https://mathai.club and upload it to OpenReview by March 15, 2026 18:00 Moscow time.

Should you have any questions regarding the program, logistics, or your presentation slot, please do not hesitate to contact us.

We look forward to your contribution to MathAI 2026.

With kind regards,

MathAI 2026 Program Committee
International Conference on Mathematics of Artificial Intelligence
https://mathai.club
OpenReview: https://openreview.net/group?id=mathai.club/MathAI/2026/Conference
Telegram: https://t.me/MathAI_club
Email: mathai.club@yandex.ru